# Antimicrobial Resistance of *Salmonella* and Characterization of Two *Mcr-1*-Harboring Isolates from Pork Products in Guangdong, China

**DOI:** 10.3390/foods14172933

**Published:** 2025-08-22

**Authors:** Zifeng Mai, Lusan Wei, Chunlei Shi, Zeqiang Zhan

**Affiliations:** 1State Key Laboratory of Microbial Metabolism, Department of Food Science & Technology, School of Agriculture & Biology, Shanghai Jiao Tong University, Shanghai 200240, China; maizifeng@sjtu.edu.cn (Z.M.); clshi@sjtu.edu.cn (C.S.); 2College of Plant Protection, South China Agricultural University, Guangzhou 510642, China; weiliusan@163.com

**Keywords:** *Salmonella*, multidrug resistance, SGI1-KI, multidrug resistance region, IncI2, colistin resistance, one health

## Abstract

*Salmonella* is a major foodborne pathogen that poses an increasing threat due to the emergence of *mcr-1*-mediated colistin resistance. However, data on *mcr-1*-positive *Salmonella* in pork products are limited. In this study, 457 samples collected in 2023 from pig slaughterhouses in Guangdong province were investigated to determine the prevalence and genomic characteristics of *mcr-1*-positive *Salmonella* isolates. We found that 92 *Salmonella* isolates (20.1%, 92/457) were recovered, representing six serotypes, including *Salmonella Typhimurium* (*n* = 29) and *Salmonella* Rissen (*n* = 29). High resistance to tetracycline (90.2%, 83/92) and multidrug resistance (58.7%, 54/92) were observed. Critically, two colistin-resistant *Salmonella Kentucky* ST198 isolates (2.2%) harboring *mcr-1* on transferable IncI2 plasmids were identified. Genomic analysis revealed a novel multidrug resistance region (MRR, ~57 kb) inserted into the *bcfH* locus (containing *floR*, *qnrS1*, *bla*_CTX-M-55_, and *aph (3’)-Ia*) and a variant *Salmonella* Genomic Island 1 (SGI1-KI, containing *tet (A)*, *sul1*, *qacEΔ1* and *aadA7*) in these isolates. The MRR and SGI1-KI may enhance bacterial survival under antibiotic selection pressure. Phylogenetic analysis showed close relatedness to human clinical strains, suggesting food chain transmission. The findings highlight a high antimicrobial resistance burden, the emergence of transferable last-resort colistin resistance (*mcr-1*), and acquisition of complex resistance determinants (MRR, SGI1-KI), underscoring an urgent need for enhanced “One Health” surveillance.

## 1. Introduction

*Salmonella* is a predominant etiological agent of foodborne gastroenteritis that imposes substantial global public health burdens [1]. Worldwide epidemiological assessments attribute approximately 131 million annual cases and 370,000 fatalities to this pathogen [2]. Surveillance systems reveal distinct regional incidence patterns: European Union data documented 65,208 confirmed human infections in 2022, equivalent to 15.3 cases per 100,000 population [3]. Concurrently, United States surveillance indicates an estimated 1.35 million yearly infections (CDC), with the FoodNet network reporting 2022 incidence at 16.3 per 100,000 [4]. In the Chinese context, *Salmonella* contamination underlies 70–80% of foodborne disease episodes [5], triggering approximately 9.87 million gastroenteritis cases annually. Nationwide foodborne gastroenteritis incidence in China is concurrently estimated at 245 cases per 100,000 population [6].

*Salmonella*, a Gram-negative bacillus within the *Enterobacteriaceae* family, currently comprises two recognized species: *Salmonella bongori* and *Salmonella enterica*, with over 2600 serotypes documented worldwide [5]. Human transmission frequently occurs through the food chain, with swine and pork products being significant vehicles [5,7]. Indeed, pork is consistently identified as a major contributor to human *Salmonella* infections [8], with epidemiological studies linking approximately 15 to 23% of human salmonellosis cases directly to pork consumption [9]. Concurrently, concerns persist regarding the extensive and often inappropriate application of antimicrobial agents in livestock production, a practice widely believed to drive the proliferation of multidrug-resistant (MDR) bacterial pathogens [10]. The rise in MDR *Salmonella* isolates poses substantial challenges for effective clinical management of infections [6]. Antimicrobial resistance (AMR) is projected to become a leading global health threat, potentially responsible for 10 million annual deaths by 2050 [11], solidifying antimicrobial-resistant bacteria as a paramount public health concern of the 21st century [12].

Colistin, a last-resort antimicrobial agent derived from bacteria, is critical for treating infections caused by Gram-negative pathogens [13]. A significant development occurred in 2015 with the discovery of the first plasmid-borne polymyxin resistance gene, *mcr-1*, in *Enterobacteriaceae* from food animals. This finding highlighted the potential for horizontal dissemination of colistin resistance among bacteria [14]. Subsequently, *mcr-1* has been identified globally in diverse bacterial species, including *Salmonella* isolates from humans, animals, food, and environmental sources, creating a complex genetic reservoir for its evolution [15]. Multiple variants of the *mcr* gene family (*mcr-1* to *mcr-5* and *mcr-9*) have now been reported in *Salmonella* across these reservoirs, with *mcr-1* being the predominant type [16]. These genes encode phosphoethanolamine transferases that modify lipid A by adding phosphoethanolamine. This enzymatic alteration reduces the net negative charge of the bacterial outer membrane, diminishing colistin binding affinity and conferring resistance [17]. Although *mcr-1*-positive *Salmonella* isolates have been detected in humans and food animals, their prevalence remains low [18]. In China, reports of foodborne bacteria harboring *mcr-1* primarily involve *Escherichia coli* (*E. coli*) from retail meats [18]; however, data concerning pork products are still limited.

Guangdong Province ranks among China’s largest pork producers and consumers, with pork products serving as a critical transmission vehicle for *Salmonella*. Currently, few studies provide comprehensive AMR surveillance data for Salmonella in pork slaughterhouses in this high-risk region. To address this knowledge gap, this study assessed the prevalence and antimicrobial susceptibility of *Salmonella* isolates obtained from pork products in Guangdong slaughterhouses in 2023. All isolates were screened via polymerase chain reaction (PCR) for *mcr* genes, and isolates carrying these genes underwent subsequent genomic characterization. The findings aim to provide foundational data to support the development of interventions designed to reduce the spread of AMR in foodborne *Salmonella.*

## 2. Materials and Methods

### 2.1. Sample Collection and Salmonella Isolation

Between January 2023 and December 2023, a total of 457 pig carcass samples were collected from two slaughterhouses (Slaughterhouse 1 and Slaughterhouse 2) in Guangdong Province, China. This study was conducted with formal approval from the operational management departments of both participating slaughterhouses. Sampling was conducted across eight months (January, March, May, June, July, August, October, December). The monthly sample distribution per slaughterhouse was highly consistent: Slaughterhouse 1 contributed 206 samples (typically 26 samples/month, reducing to 25 in October and December), while Slaughterhouse 2 contributed 251 samples (typically 31 samples/month, increasing to 32 in August, October, and December). This resulted in a consistent total of 57 samples per month for most months, except August, which had 58 samples (Appendix A). *Salmonella* isolation was performed using a modified adaptation of the ISO 6579-1:2017 protocol [19]. Pork products (carcass samples) were taken before cooling of the pig carcasses. Each sample was randomly and aseptically collected, sealed in sterile bags, and immediately stored in portable cooling boxes at 4–8 °C. All samples were transported to the laboratory within 8 h post-collection for processing. Upon arrival, approximately 25 g of each sample was rinsed with 225 mL of buffered peptone water (Huankai, China) as the initial processing step. This mixture was then subjected to shaking incubation at 37 ± 1 °C (160 rpm) for 4 h. Subsequently, 1 mL of the resulting pre-enrichment Culture was inoculated into 9 mL of tetrathionate broth (Huankai, China) and incubated at 42 °C. Following enrichment, a loopful of the culture was plated onto xylose lysine tergitol 4 agar (XLT-4; Huankai, China). Presumptive *Salmonella* colonies identified on XLT-4 were selected for confirmation via PCR targeting the *invA* gene and serotype determination was conducted using the White–Kauffmann method [20].

### 2.2. Salmonella Genomic DNA Extraction

Total genomic DNA was extracted from fresh overnight *Salmonella* cultures using the TIANamp Bacteria DNA Kit (Tiangen Biotech, Beijing, China), following the manufacturer’s protocol with minor strain-specific optimizations. Approximately 3–5 mL of culture was centrifuged (12,000 rpm, 1 min) to pellet cells and discard supernatant. Resuspend the pellet in 200 μL Buffer GA. Add 20 μL Proteinase K and 220 μL Buffer GB, mix, then incubate at 70 °C for 10 min until the solution clears. Add 220 μL ethanol, mix, and load the mixture onto the CB3 spin column. Centrifuge (12,000 rpm, 30 s) and discard flow-through. Wash the column sequentially with 500 μL Buffer GD and twice with 600 μL Buffer PW (centrifuging 30 s each time). Dry the column by centrifugation (2 min) and air-drying (2–5 min). Elute DNA with 50–200 μL Buffer TE (centrifuge 2 min). DNA concentration and quality were assessed using the NanoDrop (Thermo Fisher Scientific, Waltham, MA, USA), and samples were stored at –20 °C until downstream analysis.

### 2.3. Multi-Locus Sequence Typing of Salmonella Isolates

To determine the genetic lineages of the 92 *Salmonella* isolates, multi-locus sequence typing (MLST) was performed by sequencing seven distinct housekeeping genes (*aroC*, *dnaN*, *hemD*, *hisD*, *purE*, *sucA*, and *thrA*) [21]. PCR amplification of seven housekeeping genes was conducted in 50 μL reaction volumes containing: 25 μL of 2× Taq Master Mix (Vazyme, Nanjing), 2 μL of each primer (10 μM), 2 μL template DNA (~50 ng), and 19 μL nuclease-free water. Thermocycling conditions comprised initial denaturation at 95 °C (5 min); 30 cycles of 95 °C (30 s), 55 °C (30 s), and 72 °C (1 min); and final extension at 72 °C (10 min). Amplicons were verified by 1.0% agarose gel electrophoresis, purified, and bidirectionally sequenced. Sequence types (STs) were assigned using the *Salmonella* MLST database (EnteroBase), and phylogenetic relationships were analyzed in GrapeTree v1.5.0 to generate a minimum spanning tree based on allelic profiles [22].

### 2.4. Antimicrobial Susceptibility Testing (AST) of Salmonella Isolates

The antimicrobial susceptibility profiles of the *Salmonella* isolates were determined using both agar dilution and broth dilution methods, following Clinical and Laboratory Standards Institute (CLSI) guidelines (2023) [23]. Testing encompassed 18 antimicrobial agents spanning 10 distinct classes: ciprofloxacin (CIP), ofloxacin (OFX), nalidixic acid (NAL), cefepime (FEP), cefotaxime (CTX), ampicillin (AMP), meropenem (MEM), tigecycline (TGC), tetracycline (TET), kanamycin (KAN), amikacin (AMK), gentamicin (GEN), streptomycin (STR), sulfisoxazole (SUL), fosfomycin (FOS), chloramphenicol (CHL), azithromycin (AZI), and colistin (COL). For agar dilution, Mueller–Hinton agar (Oxoid) plates were prepared with serial two-fold dilutions of antimicrobials. Antimicrobial stocks were prepared following CLSI (2023) recommendations and stored at −80 °C prior to incorporation into agar [23]. Bacterial suspensions adjusted to 0.5 McFarland standard (1–5 × 10^8^ CFU/mL) were diluted 1:10 in sterile saline to achieve 1–5 × 10^4^ CFU/spot. Suspensions were inoculated onto plates and incubated at 37 °C for 16–20 h. Broth microdilution utilized cation-adjusted Mueller–Hinton broth (Oxoid) in commercially pre-coated 96-well plates (Thermo Fisher). Inocula were standardized to 0.5 McFarland (1–5 × 10^8^ CFU/mL) in saline, then diluted 1:100 in broth to achieve 5 × 10^5^ CFU/mL. A volume of 100 μL was dispensed per well. Plates were incubated at 37 °C for 16–20 h. Minimum inhibitory concentrations (MICs) were interpreted according to CLSI guidelines (2023) [23]. *E. coli* ATCC 25922 served as the quality control strain. Isolates exhibiting resistance to antimicrobials from three or more distinct classes were classified as MDR [24].

### 2.5. Mcr Gene Screening

Screening for *mcr* gene variants (*mcr-1* to *mcr-10*) in the *Salmonella* isolates was conducted via multiplex PCR according to established protocols with some modifications [25]. PCR (25 µL total volume) contained 12.5 µL of Taq PCR Master Mix (Vazyme, Nanjing, China), 1 µM of each primer, and ~50 ng template DNA. Amplification was conducted in a thermal cycler with initial denaturation at 95 °C (5 min); 35 cycles of 95 °C (30 s), 50~55 °C (30 s), 72 °C (1 min); and final extension at 72 °C (10 min). PCR products were electrophoresed on 1.0% agarose gels and visualized under UV. Each run included positive controls (*mcr*-carrying strains), negative controls (*mcr*-negative strains), and no-template controls. The isolates carrying *mcr* underwent whole-genome sequencing (WGS).

### 2.6. Whole-Genome Sequencing of Mcr-1-Positive Salmonella Isolates

WGS of the *mcr*-positive isolates was performed on the Illumina platform (150 bp paired-end reads). For the *mcr-1*-positive isolate Sal_P23040, complete genome assembly was achieved through long-read sequencing on the PacBio platform using ≥100× coverage. Plasmid replicons and antimicrobial resistance genes (ARGs) were characterized using the web-based tools PlasmidFinder 2.1 and ResFinder 4.1 (https://www.genomicepidemiology.org/ (accessed on 9 May 2025)). The origin of transfer (*oriT*) for mobilized genes was predicted with oriTfinder (https://bioinfo-mml.sjtu.edu.cn/oriTfinder/ (accessed on 9 May 2025)). Comparative genomic visualization of the *mcr-1* genetic context employed Easyfig 2.1 [26] and BRIG 0.95 [27].

### 2.7. Conjugation Experiments of the Mcr-1 Gene

Conjugative transfer experiments were conducted using *mcr-1*-positive *Salmonella* isolates as donor isolates and the plasmid-free, rifampicin-resistant *E. coli* J53 as the recipient. Overnight cultures were washed in phosphate-buffered saline (PBS), adjusted to OD_600_ = 0.5 (~1 × 10^8^ CFU/mL), and mixed at donor:recipient ratios of 1:1 and 1:10. Cell mixtures (200 μL) were deposited onto 0.22-μm nitrocellulose filters placed on Mueller–Hinton agar and incubated at 37 °C for 18 h. Following mating, the transconjugants were screened on MacConkey agar supplemented with rifampicin (200 mg/L) and colistin (2 mg/L). Transfer efficiency was quantified as the transconjugant count per recipient cell. Putative *mcr*-bearing transconjugants underwent PCR verification following established methodology [28]. Colistin MICs for both the parental *E. coli* J53 strain and transconjugants were determined using the AST protocol described previously [23].

### 2.8. Phylogenetic Tree Analysis of Mcr-1-Positive Salmonella Isolates and IncI2 Plasmids

To elucidate the genetic relatedness of *mcr-1*-positive *Salmonella* isolates, phylogenetic reconstruction was performed using the kSNP4.1 pipeline [29]. This analysis incorporated the isolates from this study alongside the publicly available genomes retrieved from NCBI and Enterobase. Additionally, a comparative analysis of *mcr-1*-positive IncI2 plasmids was similarly conducted, integrating one plasmid identified in this study with 14 publicly accessible plasmid sequences from NCBI. All resultant phylogenetic trees underwent visualization through the iTOL v6 online platform (https://itol.embl.de (accessed on 25 May 2025)). A complete list of all bacterial isolates and plasmids used for phylogenetic analyses is provided in Appendix A (titled ‘Isolates and plasmids for phylogenetic analysis’).

### 2.9. Data Availability Statement

The whole-genome sequences of the two *mcr-1*-positive *Salmonella* isolates have been deposited in GenBank under BioProject PRJNA1299413, with individual Biosample accessions SAMN50301257 and SAMN50301771. The accession number of Sal_P23041 is JBQLIL000000000.

### 2.10. Statistical Analysis

Statistical differences in MDR prevalence across *Salmonella* serotypes were evaluated using Pearson’s chi-square test, with significance defined at *p* < 0.05.

## 3. Results

### 3.1. Prevalence Characteristics of Salmonella Isolates from Pork Product Samples

Overall, it is shown in Table 1 and Figure 1 that 92 *Salmonella* isolates were recovered from 457 pork product samples collected between January and December 2023 across Guangdong Province, China, yielding a 20.1% (92/457) positivity rate. Notably, slaughterhouse surveillance revealed significantly elevated *Salmonella* prevalence in Slaughterhouse 1 (26.2%, 54/206) compared to Slaughterhouse 2 (15.1%, 38/251) (*p* < 0.05).

Serotyping via the White–Kauffmann method identified six distinct serovars among the isolates. *Salmonella Typhimurium* (*S. Typhimurium*) (*n* = 29) and *Salmonella Rissen* (*S. Rissen*) (*n* = 29) emerged as predominant serovars, followed by *Salmonella London* (*S. London*) (*n* = 20), *Salmonella Derby* (*S. Derby*) (*n* = 11), *Salmonella Kentucky* (*S. Kentucky*) (*n* = 2), and *Salmonella Corvallis* (*S. Corvallis*) (*n* = 1). Sankey diagram analysis further demonstrated substantial serovar diversity and epidemiological distribution across pork products (Figure 1).

Appendix A displays the minimum spanning tree results, revealing seven distinct STs. ST469 (*n* = 29) emerged as the predominant type, followed by ST155 (*n* = 20), ST34 (*n* = 17), ST19 (*n* = 12), ST40 (*n* = 11), ST198 (*n* = 2), and ST1541 (*n* = 1). The analysis further demonstrated strong correlations between specific STs and *Salmonella* serotypes: ST469 with *S. Rissen*, ST155 with *S. London*, ST34 and ST19 with *S. Typhimurium*, ST40 with *S. Derby*, and ST198 with *S. Kentucky*.

### 3.2. Antimicrobial Susceptibility Analysis of Salmonella Isolates

It is shown in Table 2 that 89.1% (82/92) of *Salmonella* isolates were resistant to at least one antimicrobial class. In total, 58.7% (54/92) of isolates exhibited MDR, while 31.5% (29/92) possessed the specific ACSSuT profile (ampicillin, chloramphenicol, streptomycin, sulfisoxazole, tetracycline resistance). The most common resistance was to tetracycline (89.1%, 82/92), followed by sulfisoxazole (77.2%, 71/92), ampicillin (58.7%, 54/92), nalidixic acid (33.7%, 31/92), streptomycin (31.5%, 29/92), chloramphenicol (31.5%, 29/92), ciprofloxacin (28.3%, 26/92), and ofloxacin (28.3%, 26/92). Kanamycin and gentamicin resistance were lower (15.2% each, 14/92). Resistance rates of 5% or less were recorded for azithromycin (4.3%, 4/92), cefotaxime (2.2%, 2/92), cefepime (2.2%, 2/92), fosfomycin (2.2%, 2/92), amikacin (1.1%, 1/92), and tigecycline (1.1%, 1/92). Critically, colistin resistance was detected in 2 isolates (2.2%, 2/92). Meropenem susceptibility was observed in all isolates (100%).

Resistance to ampicillin, tetracycline, streptomycin, sulfisoxazole and chloramphenicol was commonly observed among the *Salmonella* serovars (Table 2). Notably, MDR rates varied significantly between serovars: *S. Derby* (90.9%, 10/11) and *S. Typhimurium* (89.7%, 26/29) exhibited markedly higher rates than *S. Rissen* (37.9%, 11/29) and *S. London* (20.0%, 4/20) (*p* < 0.01). Resistance patterns for specific antibiotics followed similar trends. For example, both sulfisoxazole resistance (*S. Derby*: 90.9%; *S. Typhimurium*: 93.1%) and ampicillin resistance (*S. Derby*: 90.9%; *S. Typhimurium*: 89.7%) were widespread in these serovars, contrasting with the lower levels seen in *S. London* (sulfisoxazole: 70.0%; ampicillin: 20.0%). Quinolone resistance (ofloxacin) was also notably elevated in *S. Derby* (63.6%) and *S. Typhimurium* (41.4%), while tetracycline resistance was universal in *S. Typhimurium* (100%) and both *S. Kentucky* isolates (100%). Alarmingly, both *S. Kentucky* isolates demonstrated resistance to critically important drugs, including cefotaxime, cefepime, and colistin.

The AMR profiles of the 92 *Salmonella* isolates are detailed in Appendix A. Twenty-one distinct AMR profiles were identified. Among resistant isolates, the TET-SUL profile (resistance to 2 antimicrobial classes) was found to be most prevalent (19.6%, 18/92), followed by TET-SUL-AMP-NAL-CIP-OFX (four classes; 13.0%, 12/92), TET (one class; 10.9%, 10/92), and TET-SUL-AMP-STR-CHL-KAN (five classes; 8.7%, 8/92). An increase in the number of resisted antimicrobial classes was observed: resistance to one class was exhibited by 10.9% (10/92) of isolates (exclusively TET profile), to two classes by 19.6% (18/92; TET-SUL), to three classes by 4.3% (4/92; TET-SUL-AMP), to four classes by 14.1% (13/92), to five classes by 16.3% (15/92), and to six classes by 10.9% (10/92). Furthermore, resistance to seven antimicrobial classes was detected in two isolates. Additionally, two isolates demonstrated resistance to eight antimicrobial classes. Given that this study tested susceptibility across 10 antimicrobial classes, these two isolates met the criteria for extensively drug-resistant (XDR) classification—defined as susceptibility to only 1 or 2 classes of all agents tested [24]. This finding warrants significant concern, as available therapeutic options for infections caused by such bacteria are substantially limited.

This study also conducted a comparative analysis of AMR profiles on *Salmonella* isolates recovered from Slaughterhouse 1 (*n* = 54) and Slaughterhouse 2 (*n* = 38) (Figure 2 and Appendix A). Resistance to tetracycline and sulfisoxazole was prevalent in both slaughterhouses, but significantly higher rates were observed in Slaughterhouse 1 compared to Slaughterhouse 2 (tetracycline: 92.6% vs. 84.2%, sulfisoxazole: 85.2% vs. 65.8%). Crucially, a substantially higher MDR rate was detected in Slaughterhouse 1 (70.4%) compared to Slaughterhouse 2 (42.1%). Serovar-specific divergence was evident: in slaughterhouse one, all *S. Typhimurium* (*n* = 18) and *S. Derby* (*n* = 8) isolates were confirmed as MDR and uniformly exhibited resistance to sulfisoxazole, ampicillin, and tetracycline. *S. Derby* isolates were further characterized by exceptional quinolone resistance (nalidixic acid/ofloxacin/ciprofloxacin: 87.5%), while both *S. Kentucky* isolates demonstrated near-pan-resistance. In contrast, within slaughterhouse two, although *S. Typhimurium* (*n* = 11) maintained a high MDR rate (72.7%), *S. Rissen* (*n* = 16) showed significantly lower MDR prevalence (18.8%). Furthermore, resistance to critically important antimicrobials (cefotaxime, cefepime and colistin) was observed and exclusively identified in Slaughterhouse 1 isolates.

### 3.3. Genomic Characteristics of Two Mcr-1-Positive S. Kentucky Isolates

In total, 2 of the 92 *Salmonella* isolates (2.2%) were positive for the *mcr-1* gene and exhibited phenotypic colistin resistance (MIC = 4 mg/L for both). These two *mcr-1*-positive isolates (Sal_P23040 and Sal_P23041), identified as *S. Kentucky* ST198, demonstrated extensive co-resistance to multiple antimicrobial classes (Table 3). Genomic analysis of Sal_P23040 revealed one complete chromosome and two plasmids: p1Sal_P23040 (89,313 bp, IncI) and p2Sal_P23040 (60,661 bp, IncI2). A total of nine ARGs and three mutations in quinolone resistance-determining regions (QRDRs), conferring resistance to seven antimicrobial categories, were identified. Genetic mapping confirmed that the *mcr-1* gene was located on the IncI2 plasmid, while all other resistance genes (including *bla*_CTX-M-5_, *qnrS1*, and *tet (A)*) were chromosomally encoded (Appendix A). Sal_P23041 harbored the identical set of ARGs and QRDR mutations to Sal_P23040. The phenotypic resistance profiles strongly correlated with specific resistance determinants. The plasmid-borne *mcr-1* mediated colistin resistance. *bla*_CTX-M-5_ conferred resistance to β-lactams (ampicillin, cefotaxime, cefepime). High-level quinolone resistance (nalidixic acid, ofloxacin, ciprofloxacin) resulted from the combination of *qnrS1* and QRDR mutations in *gyrA* (S83F, D87N) and *parC* (S80I), *tet (A)* mediated tetracycline resistance. The *floR* conferred resistance to chloramphenicol. The *sul1* was responsible for sulfonamide resistance (sulfisoxazole) and aminoglycoside resistance (streptomycin, kanamycin, gentamicin) was associated with multiple genes (*aac (6’)-Iaa*, *aadA7*, *aph (3’)-Ia*). Additionally, both isolates harbored the *qacEΔ1* disinfectant resistance gene. Both isolates were recovered from pig carcass samples collected at the same slaughterhouse (Slaughterhouse 1) in July 2023, indicating a common source. Their identification as *S*. Kentucky ST198, possession of an identical set of ARGs and QRDR mutations, and nearly identical resistance profiles strongly suggest clonal relatedness. Despite this high degree of similarity, Sal_P23040 exhibited resistance to tigecycline, whereas Sal_P23041 remained susceptible.

### 3.4. Phylogenetic Relationship of Mcr-1-Positive S. Kentucky Isolates and IncI2 Plasmids

To evaluate the relationship between the two *mcr*-1-positive isolates from this study and other *mcr-1*-positive *S*. Kentucky isolates globally, we constructed a phylogenetic tree comprising 14 isolates (including our two isolates and 12 *mcr-1*-positive *S*. *Kentucky* isolates retrieved from NCBI and Enterobase databases) to assess clonality (Figure 3A). Our analysis revealed that *mcr-1*-positive *S*. *Kentucky* ST198 has been recovered from diverse sources, including humans, animals, and food. The isolates formed subclade I and subclade II. Notably, within subclade II, isolates from humans and poultry originating from countries such as France, the United Kingdom, and Morocco clustered with the isolates identified in this study. The small branch distances (scale bar < 0.1) among isolates in subclade II indicate a close genetic relationship despite their diverse origins. Specifically, our isolates (Sal_P23040 and Sal_P23041) were most closely related to a clinical human isolate (traces-0lhyZjN) from France. Among all *mcr-1*-positive isolates, ARGs conferring resistance to sulfonamides (*sul1*) and quinolones (*gyrA* (S83F) and *parC* (S80I)) were the most prevalent (14/14, 100.0%), followed by *tet* (A) (13/14, 92.9%) and *bla*_TEM-1B_ (11/14, 78.6%). Importantly, *bla*_CTX-M-55_ and *qnrS1* were detected exclusively in the isolates from this study. These findings suggest a potential role for the food chain in the dissemination of *mcr-1*-positive *S*. *Kentucky* in China.

Simultaneously, a phylogenetic analysis of 15 *mcr-1*-positive IncI2 plasmids (Figure 3B) revealed diversity and multiple branches among plasmids collected from various sources (e.g., humans, food) in different countries (e.g., China, Thailand, Oman, Argentina). Notably, the plasmid MG825374.1 (60,960 bp), originating from chicken in China, clustered closely with plasmid p2Sal_P23040 from this study. Additionally, as shown in Figure 4A, most plasmids carried only the *mcr-1* gene. The *mcr-1* coding sequence in all IncI2 plasmids was located directly upstream of an open reading frame encoding a PAP2 family protein, a feature frequently associated with *mcr-1* (Figure 4B). The complete IS*Apl*1 element downstream of the *mcr-1* cassette was detected in plasmids AP017614.1, AP017619.1, AP017622.1, CP075719.1, and KX034083.1. The IncI2 plasmids from *S*. Kentucky contained genes essential for horizontal transfer, including the origin of transfer (*oriT*), a type IV secretion system (*T4SS*) cluster, and type IV coupling protein (*T4CP*) genes. This genetic potential for transfer was confirmed by conjugation assays, in which the *mcr-1*-positive IncI2 plasmids of *S*. Kentucky isolates from this study were successfully transferred to *E. coli* J53. The estimated transfer rates were (2.3 ± 0.23) × 10^−3^ for Sal_P23040 and (1.8 ± 0.14) × 10^−4^ for Sal_P23041, and the resulting transconjugants exhibited colistin resistance.

### 3.5. Molecular Characteristics of the Multidrug Resistance Region and Salmonella Genomic Island 1

Compared to the intact *bcfH* gene found in clade ST314 (reference genome GCF_002952975.1), the insertion of a multidrug resistance region (MRR) disrupted the open reading frame (ORF) of the *bcfH* gene in ST198 isolates (Figure 5A). The *bcfH* gene encodes a multifunctional oxidoreductase essential for biogenesis of the bovine colonization factor (*bcf*) fimbriae. However, it does not appear to play a significant role in *Salmonella* colonization within animal intestines [30,31].

Although previous studies indicated that the *qnrS1* gene is typically located on IncHI2 plasmids [32], a chromosome-localized *qnrS1* was identified in this study. This finding suggests that the transfer of *qnrS1* from plasmids to the chromosome contributes to its stable maintenance within *S. Kentucky*. Furthermore, *parC* (S80I) and *gyrA* (S83F, D87N) mutations were detected in the ST198 isolates. Figure 5A illustrates that in ST198 isolates, *bla*_CTX-M-55_ was located within an approximately 57 kb MRR situated downstream of the *bcfBCDEFG* gene cluster. This MRR, bounded by IS*26* elements inserted into the *bcfH* gene, harbored multiple ARGs, including *floR*, *qnrS1*, *bla*_CTX-M-55_, and *aph (3’)-Ia*, conferring resistance to chloramphenicols, quinolones, β-lactams, and aminoglycosides.

*Salmonella* Genomic Island 1 variant K (SGI1-K) was present in the *S. Kentucky* ST198 isolates examined in this study. It is shown in Figure 5B that the variant of SGI1-K was identified specifically in the ST198 lineage. Compared to the traditional SGI1-K structure (reference genome AY463797), this variant identified in ST198 isolates was designated SGI1-KI. SGI1-KI retained the genes *tet* (A) variant, *sul1*, *qacEΔ1*, *aadA7*, and *accCA5*, but lacked the *bla*_TEM-1b_, *strAB*, IS*1133*, the segment between *S023* and *resG*, and the *tnpR* genes.

## 4. Discussion

This study detected *Salmonella* in 20.1% of samples from pig slaughterhouses in Guangdong, China. This prevalence rate is higher than rates reported both domestically and internationally. Domestically, it exceeds the rate of 8.2% found in Tibet, China [33]. Internationally, it is substantially higher than rates reported for Portugal (2.0%) [34], South Korea (9.1%) [35], Brazil (2.9%) [36], and Sardinia (13.1%) [37]. Given this high contamination level and the associated risk of human transmission, comprehensive enhancements to hygiene management across the food production chain are urgently needed to reduce contamination risks.

AMR in foodborne *Salmonella* poses a serious threat to food safety. Our findings revealed a high AMR prevalence (89.1%), with the MDR rate (58.7%) aligning with recent data on *Salmonella* from Romanian pork products [38]. Resistance to TET, SUL, and AMP was predominant in the present isolates, consistent with reports from pig slaughterhouses in China and Romania [38,39], but higher than rates documented in Ugandan pork [40]. Notably, resistance to ciprofloxacin and ofloxacin (both 28.3%) exceeded levels previously recorded [38]. Given that fluoroquinolones remain first-line therapy for human salmonellosis [41], sustained national surveillance of foodborne *Salmonella* AMR in China is critical. Furthermore, this study detected two cefepime-resistant isolates and one tigecycline-resistant *Salmonella* isolate from pork products. These resistance phenotypes are exceptionally rare in foodborne *Salmonella*. Their presence highlights the critical need to strictly monitor the use of cefepime and tigecycline in pig farming, as well as the emergence and spread of resistance. Future efforts should prioritize targeted surveillance to map the epidemiology of such resistant isolates and conduct traceability studies to determine the origins and transmission pathways of cefepime and tigecycline resistance in pig-derived *Salmonella*.

Colistin serves as a critical final-resort antibiotic for combating severe infections due to MDR Gram-negative bacteria [13]. Recognizing the threat of resistance, numerous nations have prohibited its use in animal feed. Key bans include Brazil (end of 2016), Thailand (2017), China (2017), Japan (2018), followed by Malaysia, Argentina, India (2019), Indonesia (2020), and the European Union (2022) [42]. With limited novel antibiotics emerging, clinicians increasingly rely on colistin as a ‘last-line’ therapy against these challenging pathogens [43]. Regrettably, this renewed dependence has coincided with a rise in colistin resistance [44]. The spread of the *mcr-1* gene and its variants represents a significant public health and livestock industry challenge [18]. Our study detected *mcr-1*-positive *Salmonella* isolates at a rate of 2.2%. While this prevalence is substantially lower than rates reported in eggs from Qingdao (52.6%) [45], it exceeds the 0.27% found in various Chinese retail foods between 2011 and 2016 [46]. These findings indicate that *Salmonella* carrying *mcr-1* persists at concerning levels within China’s food chain.

Research has categorized the genetic context of *mcr-1* into four distinct structural types [47]. The predominant form is the 2609 bp mobile element Tn*6330*, identified as a composite transposon. This structure harbors the *mcr-1* resistance gene alongside a 765 bp open reading frame (ORF). This ORF potentially encodes a protein resembling members of the PAP2 superfamily. Tn*6330* is characteristically flanked by two copies of IS*Apl1*, an insertion sequence from the IS*30* family [48]. Analysis in the current investigation revealed that both *mcr-1*-positive isolates exhibited an *mcr-1-pap2* configuration, notably lacking the IS*Apl1* elements. This absence suggests a possible evolutionary pathway where IS*Apl1* was excised from the genetic environment of *mcr-1* in these isolates, likely through recombination events. This structural variation aligns with previous observations indicating that IS*Apl1* loss may enhance the persistence of *mcr-1* within plasmid vectors, thereby promoting broader dissemination of this critical resistance determinant [49]. Consequently, a more comprehensive understanding of *mcr-1* transmission dynamics necessitates further investigation into its precise transposition mechanisms and evolutionary origins.

Both plasmids identified in this study that harbored the *mcr-1* gene belonged to the IncI2 incompatibility group. Critically, each plasmid carried intact genetic modules essential for conjugative transfer, including the *oriT*, *T4SS* cluster, and *T4CP* genes. This finding aligns with the previous study [50], which demonstrated that such functional clusters facilitate horizontal plasmid dissemination between bacterial isolates. Their results provide a plausible explanation for the successful experimental transfer of the two *mcr-1*-positive plasmids observed here.

*Salmonella* serovars exhibit broad host adaptability, capable of infecting diverse animal species, including mammals, birds, and insects [1]. Consequently, these broad-host-range isolates can be disseminated via feces from animals or through the consumption of various common food products [1]. Phylogenomic comparison of the two isolates characterized in this study with all available *mcr-1*-positive *S*. *Kentucky* sequences from food, human, and environmental sources revealed their close relatedness. Specifically, these isolates formed a distinct cluster alongside a human isolate originating from France. Furthermore, the minimal genetic divergence observed in the phylogenetic tree branches aligns with the consistent ST results. While in vivo transmission experiments were not conducted to confirm potential pathways, these findings imply a phylogenetic proximity between pork and human *S*. Kentucky ST198 isolates. This suggests pork-borne transmission may constitute an important vehicle for *mcr-1*-positive *S*. *Kentucky* ST198. Enhanced surveillance efforts should therefore prioritize the IncI2 plasmid-mediated spread of *mcr-1* within this lineage.

Integration of the MRR into the *bcfH* locus, disrupting this gene, potentially enhances the survival of *S. Kentucky* ST198. The MRR confers resistance to multiple antibiotics: kanamycin, streptomycin, cefotaxime, chloramphenicol, and sulfisoxazole. BcfH functions as a multifunctional oxidoreductase essential for biogenesis of the *bcf* fimbriae [30]. Although the *fim* operon shows the highest expression among *Salmonella*’s 13 known fimbrial operons under both laboratory and host conditions, *bcf* expression specifically occurs within the bovine ileum [51]. Notably, the *bcf* operon does not significantly contribute to intestinal colonization in avian hosts [52]. ARGs in *Salmonella* frequently arise from selective pressures linked to antibiotic use in animal food production and veterinary medicine for growth promotion and disease control [53]. Thus, MRR-mediated *bcfH* disruption likely provides a survival advantage for *S. Kentucky* ST198 within antibiotic-intensive environments.

A novel SGI1-K variant carrying a mercury resistance module was identified in both *S. Kentucky* ST198 isolates. Compared to the SGI1-K prototype, this variant exhibited specific deletions: *bla*_TEM-1b_, *strAB*, IS*1133*, the segment between *S023* and *resG*, and the *tnpR* gene [54]. These structural changes indicate rapid intraclonal evolution and differ from patterns observed in these isolates, suggesting potential local selection pressures driving adaptation in Chinese isolates. Previous reports describe SGI1 in *S. Kentucky* ST198 as highly mosaic, with IS*26*-mediated gene acquisitions or losses generating extensive structural diversity [55]. We propose that this inherent structural flexibility within SGI1 may confer adaptive advantages to this high-risk clone

## 5. Conclusions

This study found that the AMR levels in *Salmonella* from Chinese pork products were alarmingly high in 2023, with an MDR rate of 58.7% and a complex, diverse resistance profile. Resistance to critically important clinical drugs (e.g., tigecycline) used for MDR *Salmonella* infection treatment was confirmed. Notably, IncI2 plasmids carrying the *mcr-1* gene were identified in *S. Kentucky* ST198 isolates from pork products. Horizontal transfer of these plasmids constitutes a key dissemination vehicle for *mcr-1* among pork-derived *Salmonella* in China. Phylogenetic evidence further indicates that pork-borne transmission facilitates the spread of *mcr-1*-positive *S. Kentucky* ST198 from the food chain to humans. The MRR and the SGI1-KI, containing multiple ARGs, may enhance *Salmonella* survival under antibiotic selection pressure. Building on these concerning findings, a multi-faceted mitigation strategy is imperative. This includes strictly reducing and phasing out the use of colistin and other critically important antimicrobials as growth promoters or prophylactics in swine production to alleviate selection pressure. Concurrently, accelerating the development and deployment of effective alternatives—such as vaccines targeting prevalent *Salmonella* serovars, probiotics, bacteriophages, enhanced biosecurity, and improved husbandry practices—is crucial to decrease reliance on antimicrobials. Furthermore, exploring novel interventions targeting epidemic plasmid dissemination, like conjugation inhibitors or CRISPR-Cas-based plasmid curing, could help limit the horizontal spread of *mcr-1* and other resistance determinants. Strengthening integrated “One Health” surveillance systems across human, veterinary, and environmental sectors is essential for real-time detection of emerging resistance and enabling targeted interventions at critical points along the pork production continuum (farm, slaughter, processing, retail).

## Figures and Tables

**Figure 1 foods-14-02933-f001:**
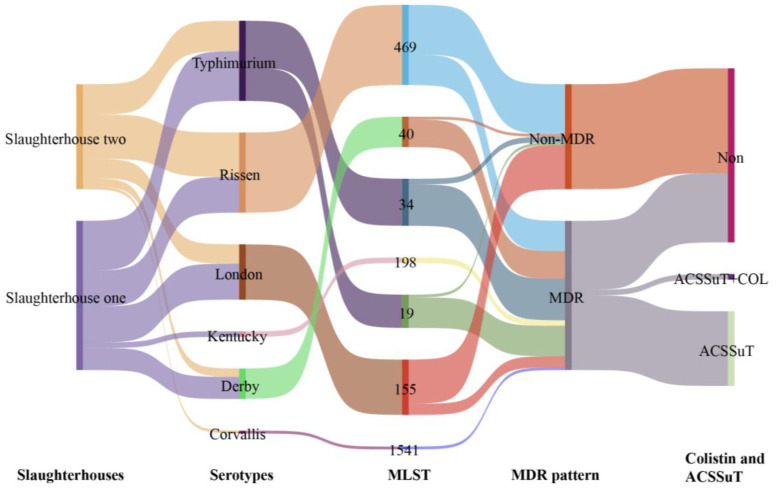
Characterization of *Salmonella* isolates (*n* = 92) from pork products: slaughterhoues; serotypes; MLST; MDR patterns (Sankey plot). Rectangle height corresponds to the number of *Salmonella* isolates. The line graph depicts serotype distribution across slaughterhouses, while distinct colors represent different serotypes for STs, MDR status, colistin resistance, and ACSSuT resistance profiles.

**Figure 2 foods-14-02933-f002:**
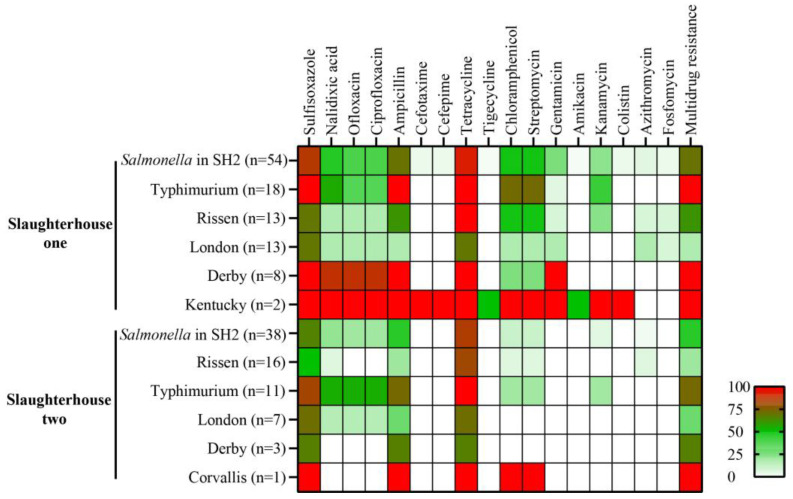
The serotype and antibiotic resistance diversity of *Salmonella* in two slaughterhouses. The average resistance distribution (in percent) of total *Salmonella* isolates and various serotypes in Slaughterhouse 1 and Slaughterhouse 2.

**Figure 3 foods-14-02933-f003:**
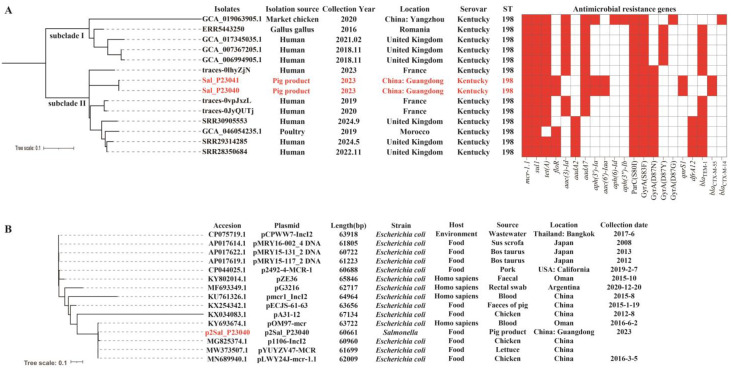
Phylogenetic relationship of *mcr-1*-positive *Salmonella* Kentucky isolates and IncI2 plasmids. (**A**) Phylogenetic analysis of 14 *mcr-1*-positive *S. Kentucky* isolates, comprising 12 isolates sourced from NCBI and Enterobase. Isolates sequenced in this study (Sal_P23040 and Sal_P23041) are denoted by red markers. The basic information of the isolates and the presence of antimicrobial resistance genes (red) are shown. (**B**) Phylogenetic analysis of 15 IncI2 plasmids carrying the *mcr-1* gene, comprising 14 plasmids sourced from NCBI. The plasmid sequenced in this study (p2Sal_P23040) is denoted by a red marker.

**Figure 4 foods-14-02933-f004:**
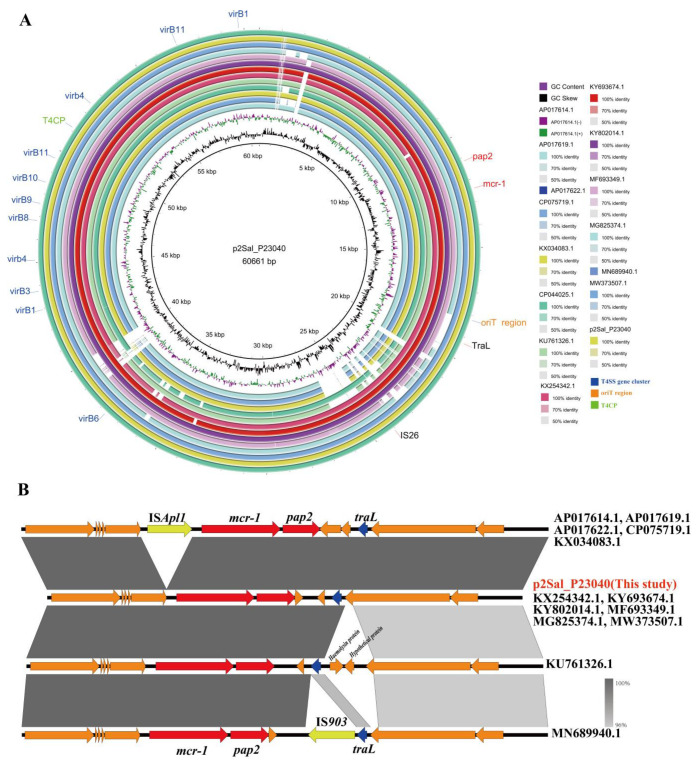
Comparative analysis of the complete IncI2 plasmids carrying the *mcr-1* gene. (**A**) Comparative analysis of 15 complete IncI2 plasmids carrying the *mcr-1* gene using the BRIG tool. The outermost ring represents the annotation of plasmid p2Sal_P23040. Genes are color-coded based on their functional annotation. Genes responsible for horizontal transfer: *T4CP* is colored green, the *T4SS* clusters are colored blue. The antimicrobial resistance genes *mcr-1* and *pap2* are colored red. (**B**) Comparison of the *mcr-1* genetic context across all plasmids. All plasmids contain the *pap2-mcr-1* unit. Plasmids AP017614.1, AP017619.1, AP017622.1, CP075719.1 and KX034083.1 contain the transposon IS*Apl1*; the other plasmids lack this transposon.

**Figure 5 foods-14-02933-f005:**
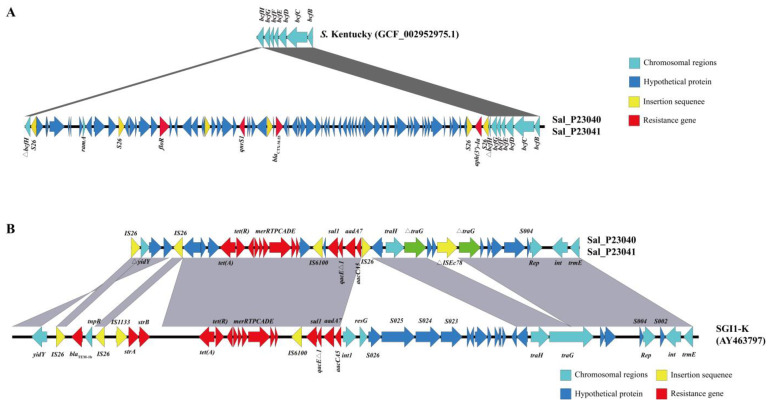
Molecular characteristics of the multidrug resistance region and *Salmonella* Genomic Island 1. (**A**) Linear comparison of the genetic structure of the multidrug resistance region (MMR) inserted into the *bcfH* gene in *S. Kentucky* ST198 isolates. Chromosomal genes are marked in light blue, while insertion sequences (IS) and resistance genes are marked in yellow and red, respectively. (**B**) Linear comparison of the genetic structure of the *Salmonella* Genomic Island 1 (SGI1) variant inserted downstream of the *trmE* gene in *S. Kentucky* ST198 isolates. Results compared with the reference SGI (AY463797). Resistance genes are marked in red as shown in the legend, and insertion sequences (IS) are marked in yellow. The gray shadowing denotes BLASTn matching regions.

**Table 1 foods-14-02933-t001:** Prevalence of *Salmonella* from pork product samples in 2023 in Guangdong, China.

Pig Slaughterhouses	Number of Samples	Number of Samples Positive for *Salmonella*	Percentage of Positive Samples (%)
One	206	54	26.2
Two	251	38	15.1
Total	457	92	20.1

**Table 2 foods-14-02933-t002:** Antimicrobial Resistance Among *Salmonella* and *Salmonella* serotypes recovered from pork products in Guangdong, China.

Antimicrobials	Number of Antimicrobial-Resistant Isolates Among (%)						
	All *Salmonella* (*n* = 92)	Typhimurium (*n* = 29)	Rissen (*n* = 29)	London (*n* = 20)	Derby (*n* = 11)	Kentucky (*n* = 2)	Corvallis (*n* = 1)
Folate pathway inhibitors							
Sulfisoxazole	71 (77.2)	27 (93.1)	17 (58.6)	14 (70.0)	10 (90.9)	2 (100)	1 (100)
Quinolones							
Nalidixic acid	31 (33.7)	16 (55.2)	3 (10.3)	3 (15.0)	7 (63.6)	2 (100)	0 (0)
Ofloxacin	26 (28.3)	12 (41.4)	2 (6.9%)	3 (15.0)	7 (63.6)	2 (100)	0 (0)
Ciprofloxacin	26 (28.3)	12 (41.4)	2 (6.9%)	3 (15.0)	7 (63.6)	2 (100)	0 (0)
β-Lactam							
Ampicillin	54 (58.7)	26 (89.7)	11 (37.9)	4 (20.0)	10 (90.9)	2 (100)	1 (100)
Cefotaxime	2 (2.2)	0 (0)	0 (0)	0 (0)	0 (0)	2 (100)	0 (0)
Cefepime	2 (2.2)	0 (0)	0 (0)	0 (0)	0 (0)	2 (100)	0 (0)
Tetracyclines							
Tetracycline	82 (89.1)	29 (100)	26 (89.7)	14 (70.0)	10 (90.9)	2 (100)	1 (100)
Tigecycline	1 (1.1)	0 (0)	0 (0)	0 (0)	0 (0)	1 (50.0)	0 (0)
Phenicols							
Chloramphenicol	29 (31.5)	15 (51.7)	7 (24.1)	2 (10.0)	2 (18.2)	2 (100)	1 (100)
Aminoglycosides							
Streptomycin	29 (31.5)	15 (51.7)	7 (24.1)	2 (10.0)	2 (18.2)	2 (100)	1 (100)
Gentamicin	14 (15.2)	1 (3.4)	1 (3.4)	2 (10.0)	8 (72.7)	2 (100)	0 (0)
Amikacin	1 (1.1)	0 (0)	0 (0)	0 (0)	0 (0)	1 (50.0)	0 (0)
Kanamycin	14 (15.2)	9 (31.0)	3 (10.3)	0 (0)	0 (0)	2 (100)	0 (0)
Polymyxins							
Colistin	2 (2.2)	0 (0)	0 (0)	0 (0)	0 (0)	2 (100)	0 (0)
Macrolides							
Azithromycin	4 (4.3)	0 (0)	2 (6.9%)	2 (10.0)	0 (0)	0 (0)	0 (0)
Fosfomycins							
Fosfomycin	2 (2.2)	0 (0)	1 (3.4)	1 (5.0)	0 (0)	0 (0)	0 (0)
Carbapenems							
Meropenem	0 (0)	0 (0)	0 (0)	0 (0)	0 (0)	0 (0)	0 (0)
Multidrug resistance	54 (58.7)	26 (89.7)	11 (37.9)	4 (20.0)	10 (90.9)	2 (100.0)	1 (100.0)

**Table 3 foods-14-02933-t003:** Antimicrobial susceptibility, conjugation rate, and whole genome analysis of two *mcr-1*-positive isolates collected in this study.

Category	Antimicrobial Class	Antimicrobial Agent	Sal_P23040		Sal_P23041	
			MIC (mg/L)	Related Genes	MIC (mg/L)	Related Genes
Antimicrobial susceptibility testing	Folate pathway inhibitors	Sulfisoxazole	>2048	*sul1*	>2048	*sul1*
	Quinolones	Nalidixic acid	>128	*qnrS1*, *gyrA* (S83F, D87N), *parC* (S80I)	>128	*qnrS1*, *gyrA* (S83F, D87N), *parC* (S80I)
		Ofloxacin	16		16	
		Ciprofloxacin	16		16	
	β-Lactam	Ampicillin	>256	*bla* _CTX-M-55_	>256	*bla* _CTX-M-55_
		Cefotaxime	32		32	
		Cefepime	16		16	
	Tetracyclines	Tetracycline	128	*tet(A)*	64	*tet(A)*
		Tigecycline	4		<0.5	
	Phenicols	Chloramphenicol	128	*floR*	128	*floR*
	Aminoglycosides	Streptomycin	128	*aac (6’)-Iaa, aadA7, aph (3’)-Ia*	128	*aac (6’)-Iaa, aadA7, aph (3’)-Ia*
		Gentamicin	16		16	
		Amikacin	64		<4	
		Kanamycin	256		256	
	Polymyxins	Colistin	4	*mcr-1.1*	4	*mcr-1.1*
	Macrolides	Azithromycin	<2		<2	
	Fosfomycins	Fosfomycin	<4		<4	
	Carbapenems	Meropenem	<0.25		<0.25	
Collection time			2023		2023	
Serotype			*Salmonella Kentucky*		*Salmonella Kentucky*	
Sequence type			ST198		ST198	
*mcr-1* location			IncI2		IncI2	
Conjugation rate			(2.3 ± 0.23) × 10^−3^		(1.8 ± 0.14) × 10^−4^	

## Data Availability

The data in the study are available from the corresponding author on reasonable request.

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
