# Peer review of "Antimicrobial Resistance of Salmonella and Characterization of Two Mcr-1-Harboring Isolates from Pork Products in Guangdong, China"

_foods, 2025, doi:10.3390/foods14172933_

Round 1

Reviewer 1 Report

Comments and Suggestions for Authors

The study aimed to assess the frequency of Salmonella in samples of pork-derived products collected from slaughterhouses, to establish the antimicrobial resistance (AMR) profile, detect genes related to colistin resistance, and characterize the genome of isolates carrying the mcr gene (associated with colistin resistance).
The topic is highly relevant, as colistin resistance in this area and in the studied products is still underexplored. Moreover, AMR epidemiological surveillance studies contribute to understanding the current status of this issue, which is of major concern to the WHO. Therefore, the study topic has scientific merit.

Remarks

  • In the keywords, replace terms that already appear in the title to improve the manuscript’s indexing.
  • In section 2.1, describe the sampled products and their quantities, or alternatively, make this information available as a supplementary file.
  • Before section 2.2, the authors should include a section describing the DNA extraction method and how its quality was assessed, prior to presenting the MLST methodology. In its current format, the sequence of analyses lacks coherence.
  • Line 106: Cite the MDR criterion (https://doi.org/10.1111/j.1469-0691.2011.03570.x).
  • Line 128: Do not specify the serotypes in the Methods section. This belongs in the Results.
  • In section 2.7, provide a supplementary file listing all isolates used for the phylogenetic analysis.
  • The section on genome deposit should be separated from statistical analyses. Presenting both together is inappropriate.
  • In Table 1, use only the slaughterhouse number; do not repeat the word “slaughterhouse” since it is already in the column header. Also, separate the “total” row.
  • Line 181: Salmonella should be italicized. Perform a double-check throughout the manuscript.
  • Line 184: The sentence is unclear—there appears to be a formatting issue caused by the insertion of a figure. Please revise.
  • Lines 194–195: Can these be considered XDR? Check the reference I suggested.
  • In section 3.3, clarify whether the isolates positive for this gene came from the same slaughterhouse, farm, or other common sources, considering their high genetic similarity. Could they be clones of one another?
  • Line 388: The term “vectors” should be used exclusively for invertebrates. Please revise.
  • The Data Availability Statement must include the genome deposit information in GenBank, along with accession numbers.

Author Response

Reviewer 1

Comments to the Author

The study aimed to assess the frequency of Salmonella in samples of pork-derived products collected from slaughterhouses, to establish the antimicrobial resistance (AMR) profile, detect genes related to colistin resistance, and characterize the genome of isolates carrying the mcr gene (associated with colistin resistance).

The topic is highly relevant, as colistin resistance in this area and in the studied products is still underexplored. Moreover, AMR epidemiological surveillance studies contribute to understanding the current status of this issue, which is of major concern to the WHO. Therefore, the study topic has scientific merit.

Response: Thank you very much for the positive comments, helpful suggestions and detailed corrections. We have reviewed the comments carefully and have revised the manuscript as your suggestions.

Question 1: In the keywords, replace terms that already appear in the title to improve the manuscript’s indexing.

Response: We sincerely thank you for this valuable suggestion. As recommended, we have revised the keywords to eliminate redundancy with the title and to better reflect the study’s novel findings. The updated keyword list is now: Salmonella; Multidrug Resistance; SGI1-KI; Multidrug Resistance Region; IncI2; Colistin Resistance; One Health. The relevant sentence “Keywords: Salmonella; Multidrug Resistance; SGI1-KI; Multidrug Resistance Region; IncI2; Colistin Resistance; One Health” was in Line 31-32.

Question 2: In section 2.1, describe the sampled products and their quantities, or alternatively, make this information available as a supplementary file.

Response: We thank your comment. In Section 2.1 (Sample collection and Salmonella isolation), we have explicitly described the sampled product as pig carcass samples and provided detailed quantitative information:

  • Total samples: 457
  • Source: Slaughterhouse 1 (n = 206), Slaughterhouse 2 (n = 251)
  • Sampling months: January, March, May, June, July, August, October, December (8 months)

4) Monthly distribution:

(1) Slaughterhouse 1: Typically 26/month (25 in Oct/Dec)

(2) Slaughterhouse 2: Typically 31/month (32 in Aug/Oct/Dec)

(3) Total/month: 57 samples (58 in August)

Additionally, the complete monthly dataset is provided in Table S1 in the supplementary file.

We have added the relevant sentence “Between January 2023 and December 2023, a total of 457 pig carcass samples were collected from two slaughterhouses (Slaughterhouse 1 and Slaughterhouse 2) in Guangdong Province, China. This study was conducted with formal approval from the operational management departments of both participating slaughterhouses. Sampling was conducted across eight months (January, March, May, June, July, August, October, December). The monthly sample distribution per slaughterhouse was highly consistent: Slaughterhouse 1 contributed 206 samples (typically 26 samples/month, reducing to 25 in October and December), while Slaughterhouse 2 contributed 251 samples (typically 31 samples/month, increasing to 32 in August, October, and December). This resulted in a consistent total of 57 samples per month for most months, except August which had 58 samples (Table S1).” in Line 87-96.

Question 3: Before section 2.2, the authors should include a section describing the DNA extraction method and how its quality was assessed, prior to presenting the MLST methodology. In its current format, the sequence of analyses lacks coherence.

Response: We sincerely appreciate your valuable suggestion to enhance the methodological coherence of our manuscript. As recommended, we have now added a dedicated section describing the DNA extraction protocol and quality assessment before Section 2.2 (MLST analysis).

The title of the new Section is 2.2 Salmonella Genomic DNA Extraction. We have added the relevant sentence “Total genomic DNA was extracted from fresh overnight Salmonella cultures using the TIANamp Bacteria DNA Kit (Tiangen Biotech, Beijing, China), following the manufacturer’s protocol with minor strain-specific optimizations. Approximately 3–5 mL of culture was centrifuged (12,000 rpm, 1 min) to pellet cells and discard supernatant. Resuspend the pellet in 200 μl Buffer GA. Add 20 μl Proteinase K and 220 μl Buffer GB, mix, then incubate at 70℃ for 10 min until the solution clears. Add 220 μl ethanol, mix, and load the mixture onto the CB3 spin column. Centrifuge (12,000 rpm, 30 sec) and discard flow-through. Wash the column sequentially with 500 μl Buffer GD and twice with 600 μl Buffer PW (centrifuging 30 sec each time). Dry the column by centrifugation (2 min) and air-drying (2–5 min). Elute DNA with 50–200 μl Buffer TE (centrifuge 2 min). DNA concentration and quality were assessed using the NanoDrop (Thermo Fisher Scientific), and samples were stored at –20℃ until downstream analysis.” in Line 108-119.

Question 4: Line 106: Cite the MDR criterion (https://doi.org/10.1111/j.1469-0691.2011.03570.x).

Response: We sincerely thank you for highlighting this important reference. As suggested, we have now cited the MDR classification criteria proposed by Magiorakos et al. (2012) in our revised manuscript. This citation appears in Line 163 of the revised manuscript, where we define multidrug-resistant (MDR) Salmonella isolates in accordance with the international consensus guidelines.

Question 5: Line 128: Do not specify the serotypes in the Methods section. This belongs in the Results.

Response: We sincerely appreciate this valuable observation. As rightly noted, the specification of serotypes belongs in the Results section rather than Methods. Accordingly, we have removed the S. Kentucky from Line 193 of the Methods section. This information will now be presented exclusively in the Results section where serotype distributions are formally reported. We have changed the sentence “Phylogenetic tree analysis of mcr-1-positive S. Kentucky isolates and IncI2 plasmids” to “Phylogenetic tree analysis of mcr-1-positive Salmonella isolates and IncI2 plasmids” in Line 194

Question 6: In section 2.7, provide a supplementary file listing all isolates used for the phylogenetic analysis.

Response: We thank you for this constructive suggestion. As requested, we have now provided a supplementary file listing all isolates and plasmids used for the phylogenetic analysis in Section 2.8. This file (designated Supplementary File S2: ‘Isolates and plasmids for phylogenetic analysis).  As recommended, we have added the relevant sentence “A complete list of all bacterial isolates and plasmids used for phylogenetic analyses is provided in Supplementary File S2 (titled ‘Isolates and plasmids for phylogenetic analysis’).” In Line 201-203.

Question 7: The section on genome deposit should be separated from statistical analyses. Presenting both together is inappropriate.

Response: We sincerely appreciate this valuable suggestion regarding the structural organization of our manuscript. As recommended, we have now separated the genome deposit information from statistical analyses by creating a dedicated standalone section for sequence data submission. Specifically: The original Section 2.9 has been restructured into two distinct sections:

New Section 2.9: Data availability statement (focusing exclusively on database accession details) in Line 215.

Revised Section 2.10: Statistical analyses (retaining methodological descriptions) in Line 220.

This modification ensures logical compartmentalization of methodological reporting and fully aligns with academic conventions.

Question 8: In Table 1, use only the slaughterhouse number; do not repeat the word “slaughterhouse” since it is already in the column header. Also, separate the “total” row.

Response: We sincerely appreciate your careful review of Table 1. We have implemented the following revisions according to your suggestions:

  • Removed redundant labels: The term "slaughterhouse" has been deleted from all entries under the Slaughterhouse column header (now simply labeled "One" and "Two" as requested). The relevant data was in Line 240-241
  • Optimized table clarity: The revised table now maintains the merged "Total" row at the bottom. We found that separating this row with horizontal lines significantly reduced visual coherence in our complex data structure, potentially causing confusion when interpreting multi-category counts.

The current format ensures the immediate distinction between slaughterhouse-specific data and totals. Should you feel further adjustments are needed, we would be glad to provide alternative formats.

Question 9: Line 181: Salmonella should be italicized. Perform a double-check throughout the manuscript.

Response: We sincerely thank you for highlighting this important formatting requirement. We have now corrected Salmonella to italicized form in Line 271 and conducted a full-text review of the manuscript to ensure the correct writing of Salmonella.

Question 10: Line 184: The sentence is unclear—there appears to be a formatting issue caused by the insertion of a figure. Please revise.

Response: We sincerely appreciate your feedback on the formatting issues caused by figure insertion. We have revised the ambiguous sentence and adjusted the text layout to ensure logical flow around figures—specifically, images originally embedded within paragraphs have now been placed beneath the relevant paragraphs. Clarity throughout the revised manuscript has been verified. Should further refinement be needed, we would be glad to incorporate additional suggestions. The relevant sentence was in Line 253-266.

Question 11: Lines 194–195: Can these be considered XDR? Check the reference I suggested.

Response: Thank you very much for your valuable suggestion and for bringing this important point to our attention. We carefully reviewed the definition of extensively drug-resistant (XDR) organisms as outlined in the reference you recommended (Magiorakos et al., Clin Microbiol Infect, 2012). As you rightly noted, our study tested susceptibility across 10 antimicrobial categories. Upon thorough re-examination, we confirmed that two isolates meet the XDR criteria defined in this reference. Both isolates demonstrated resistance to 8 out of the 10 tested antimicrobial categories, meaning they remain susceptible to agents in only 1–2 categories, consistent with the standard XDR definition (non-susceptibility to ≥1 agent in all but ≤2 categories).

We have now explicitly cited this reference (Magiorakos et al., Clin Microbiol Infect, 2012) in Line 285 in the revised manuscript to support the XDR classification of these isolates. We sincerely appreciate your rigorous review, which has enhanced the precision of our resistance profiling and strengthened the manuscript.

We have added the relevant sentence “Furthermore, resistance to 7 antimicrobial classes was detected in two isolates. Additionally, two isolates demonstrated resistance to 8 antimicrobial classes. Given that this study tested susceptibility across 10 antimicrobial classes, these two isolates met the criteria for extensively drug-resistant (XDR) classification—defined as susceptibility to only 1 or 2 classes of all agents tested. This finding warrants significant concern, as available therapeutic options for infections caused by such bacteria are substantially limited.” in Line 281-287.

Question 12:  In section 3.3, clarify whether the isolates positive for this gene came from the same slaughterhouse, farm, or other common sources, considering their high genetic similarity. Could they be clones of one another?

Response: Thank you for this important question. We have now clarified this point in the revised manuscript text (Section 3.3) and provide the details below:

  • Common Source: As stated in the original paragraph, both mcr-1-positive isolates, Sal_P23040 and Sal_P23041, originated from the same slaughterhouse (Slaughterhouse 1) and were recovered from pig carcass samples in the same time (July 2023).
  • Genetic Similarity & Clonal Relatedness: The isolates were identified as the same sequence type ( Kentucky ST198). Genomic analysis revealed they harbored an identical set of antimicrobial resistance genes and identical mutations in the quinolone resistance-determining regions. Furthermore, the resistance profiles were nearly identical phenotypically.
  • Minor Difference: The only phenotypic difference noted was resistance to tigecycline in Sal_P23040 but not Sal_P23041.

We have added the relevant sentence “Both isolates were recovered from pig carcass samples collected at the same slaughterhouse (Slaughterhouse 1) in July 2023, indicating a common source. Their identification as S. Kentucky ST198, possession of an identical set of ARGs and QRDR mutations, and nearly identical resistance profiles strongly suggest clonal relatedness. Despite this high degree of similarity, Sal_P23040 exhibited resistance to tigecycline, whereas Sal_P23041 remained susceptible.” in Line 327-332.

Question 13: Line 388: The term “vectors” should be used exclusively for invertebrates. Please revise.

Response: Thank you very much for your suggestion. As recommended, we have replaced the term "vectors" with "vehicles" in Line 484.

Question 14: The Data Availability Statement must include the genome deposit information in GenBank, along with accession numbers.

Response: We sincerely appreciate your guidance on enhancing the Data Availability Statement. We have now explicitly incorporated the genome deposit details into this section, stating: "The whole-genome sequences of the two mcr-1-positive Salmonella isolates have been deposited in GenBank under BioProject PRJNA1299413, with individual Biosample accessions SAMN50301257 and SAMN50301771. The accession number of Sal_P23041 is JBQLIL000000000" In Line 216-219. This revision ensures full compliance with genomic data sharing standards.

Reviewer 2 Report

Comments and Suggestions for Authors

The manuscript entitled: “Antimicrobial Resistance of Salmonella and Characterization of Two mcr-1-harboring Isolates from Pork Products in Guangdong, China”, presents a study on Salmonella found in samples of Chinese pork products in 2023.  The scope and rationale of the proosed study should be better cleared. The study evidences also the relevant issue of antimicrobial resistance. It should be cleared by the Authors why the reported data refer only to the year 2023 and not also to the following years: this would be useful to give a better overall assessment of the problem of this contaminant in pork meat. Some comments are reported in the following. It wuold be useful to mention which “specimens” have been analyzed (see line 80) and the procedure used. The study has been approved by some Authority? Please indicate this also. The experimental procedures followed should be more detailed (e.g. see paragraph 2.3, etc.) for better clarity. The Authors conclude that the salmonella levels in Chinese pork products were very high in 2023, nonethelss also more recent data would be needed to give a wider overall picture of this severe issue. Morevoer a comparison with similar situations, if observed in other Countries, would be useful to give a better assessment of the problem and to counteract it. At the same time, in the Conclusion section, a perspective possible way to find a solution or possible alternatives should be added by the Authors after the data which they reported.

Comments on the Quality of English Language

The manuscript entitled: “Antimicrobial Resistance of Salmonella and Characterization of Two mcr-1-harboring Isolates from Pork Products in Guangdong, China”, presents a study on Salmonella found in samples of Chinese pork products in 2023. The study evidences also the relevant issue of antimicrobial resistance. It should be cleared by the Authors why the reported data refer only to the year 2023 and not also to the following years: this would be useful to give a better overall assessment of the problem of this contaminant in pork meat. Some comments are reported in the following. It wuold be useful to mention which “specimens” have been analyzed (see line 80) and the procedure used. The study has been approved by some Authority? Please indicate this also. The experimental procedures followed should be more detailed (e.g. see paragraph 2.3, etc.) for better clarity. The Authors conclude that the salmonella levels in Chinese pork products were very high in 2023, nonethelss also more recent data would be needed to give a wider overall picture of this severe issue. Morevoer a comparison with similar situations, if observed in other Countries, would be useful to give a better assessment of the problem and to counteract it. At the same time, in the Conclusion section, a perspective possible way to find a solution or possible alternatives should be added by the Authors after the data which they reported.

Author Response

Reviewer 2

Comments to the Author

The manuscript entitled: “Antimicrobial Resistance of Salmonella and Characterization of Two mcr-1-harboring Isolates from Pork Products in Guangdong, China”, presents a study on Salmonella found in samples of Chinese pork products in 2023. The scope and rationale of the proosed study should be better cleared. The study evidences also the relevant issue of antimicrobial resistance. It should be cleared by the Authors why the reported data refer only to the year 2023 and not also to the following years: this would be useful to give a better overall assessment of the problem of this contaminant in pork meat. Some comments are reported in the following. It wuold be useful to mention which “specimens” have been analyzed (see line 80) and the procedure used. The study has been approved by some Authority? Please indicate this also. The experimental procedures followed should be more detailed (e.g. see paragraph 2.3, etc.) for better clarity. The Authors conclude that the salmonella levels in Chinese pork products were very high in 2023, nonethelss also more recent data would be needed to give a wider overall picture of this severe issue. Morevoer a comparison with similar situations, if observed in other Countries, would be useful to give a better assessment of the problem and to counteract it. At the same time, in the Conclusion section, a perspective possible way to find a solution or possible alternatives should be added by the Authors after the data which they reported.

Response: Thank you very much for the positive comments, helpful suggestions, detailed corrections and questions. We have reviewed the comments carefully and will address them point by point.

Question 1: The scope and rationale of the proposed study should be better cleared.

Response: Thank you very much. This study addresses a critical gap: despite Guangdong being China's largest pork producer/consumer and pork being a key Salmonella vector, comprehensive AMR data, especially for mobile colistin resistance (mcr) genes, from its slaughterhouses was limited. To fill this gap, we specifically assessed Salmonella in pork from Guangdong slaughterhouses (2023). Our scope included: determining prevalence, conducting phenotypic AMR testing on all isolates, PCR screening all isolates for mcr-1 to mcr-10, and performing whole-genome sequencing on mcr-positive isolates for genomic characterization. This provides essential foundational data to support interventions curbing AMR spread, particularly concerning colistin resistance, in this high-risk foodborne reservoir.

We have added the relevant sentence “Guangdong Province ranks among China's largest pork producers and consumers, with pork products serving as a critical transmission vehicle for Salmonella. Currently, few studies provide comprehensive AMR surveillance data for Salmonella in pork slaughterhouses in this high-risk region. To address this knowledge gap, this study assessed the prevalence and antimicrobial susceptibility of Salmonella isolates obtained from pork products in Guangdong slaughterhouses in 2023. All isolates were screened via polymerase chain reaction (PCR) for mcr genes, and isolates carrying these genes underwent subsequent genomic characterization. The findings aim to provide foundational data to support the development of interventions designed to reduce the spread of AMR in foodborne Salmonella.” in Line 76-84.

Question 2: The study evidences also the relevant issue of antimicrobial resistance. It should be cleared by the Authors why the reported data refer only to the year 2023 and not also to the following years: this would be useful to give a better overall assessment of the problem of this contaminant in pork meat.

Response: We sincerely thank you for this valuable perspective. While multi-year data would indeed enhance our understanding of Salmonella contamination dynamics – particularly for mcr-1-positive isolates – this study's primary objective was to establish critical baseline data for Guangdong slaughterhouses specifically in 2023. Comprehensive sample processing, antimicrobial susceptibility testing, and whole-genome sequencing proved highly resource-intensive. Consequently, we focused exclusively on the 2023 monitoring cycle to ensure both the timely reporting of pivotal findings (notably the identification of mcr-1-positive ST198 isolates harboring complex resistance islands) and the thorough analysis of all 457 samples within project constraints. Expanding to multi-year surveillance would require substantial additional resources beyond this project's scope. We therefore limited intensive analysis to the 2023 dataset, while fully acknowledging the importance of longitudinal studies should future resources become available.

Question 3: It would be useful to mention which “specimens” have been analyzed (see line 80) and the procedure used.

Response: The specimens analyzed in this study were pig carcass samples. Pig carcass samples were collected aseptically immediately after slaughter and prior to cooling. Each carcass sample was randomly collected, sealed in sterile bags, and stored in portable cooling boxes at 4–8°C. All samples were transported to the laboratory and processed within 8 hours post-collection. Upon arrival, approximately 25 g of each sample was rinsed with 225 mL of buffered peptone water.

We have added the relevant sentence “Between January 2023 and December 2023, a total of 457 pig carcass samples were collected from two slaughterhouses (Slaughterhouse 1 and Slaughterhouse 2) in Guangdong Province, China. This study was conducted with formal approval from the operational management departments of both participating slaughterhouses. Sampling was conducted across eight months (January, March, May, June, July, August, October, December). The monthly sample distribution per slaughterhouse was highly consistent: Slaughterhouse 1 contributed 206 samples (typically 26 samples/month, reducing to 25 in October and December), while Slaughterhouse 2 contributed 251 samples (typically 31 samples/month, increasing to 32 in August, October, and December). This resulted in a consistent total of 57 samples per month for most months, except August which had 58 samples (Table S1). Salmonella isolation was performed using a modified adaptation of the ISO 6579-1:2017 protocol. Pork products (carcass samples) were taken before cooling of the pig carcasses. Each sample was randomly and aseptically collected, sealed in sterile bags, and immediately stored in portable cooling boxes at 4–8℃. All samples were transported to the laboratory within 8 hours post-collection for processing. Upon arrival, approximately 25 g of each sample was rinsed with 225 mL of buffered peptone water (Huankai, China) as the initial processing step” in Line 87-101.

Question 4: The study has been approved by some Authority? Please indicate this also.

Response: Yes, this study was conducted with formal approval from the operational management departments of both participating slaughterhouses. These departments hold the authority within their respective facilities for overseeing operations, including procedures related to animal handling and welfare assessments conducted on-site. Their approval was obtained prior to the commencement of the study activities within each slaughterhouse.

We have added the relevant sentence “This study was conducted with formal approval from the operational management departments of both participating slaughterhouses.” in Line 89-90.

Question 5: The experimental procedures followed should be more detailed (e.g. see paragraph 2.3, etc.) for better clarity.

Response: We thank you for this valuable suggestion to enhance the clarity of our manuscript. We agree that a more detailed description of the experimental procedures is crucial for reproducibility and understanding. In response to this comment, we have significantly expanded the descriptions of the experimental procedures throughout the Methods section.

  • We have added a new Section “2.2 Salmonella Genomic DNA Extraction. Total genomic DNA was extracted from fresh overnight Salmonella cultures using the TIANamp Bacteria DNA Kit (Tiangen Biotech, Beijing, China), following the manufacturer’s protocol with minor strain-specific optimizations. Approximately 3–5 mL of culture was centrifuged (12,000 rpm, 1 min) to pellet cells and discard supernatant. Resuspend the pellet in 200 μl Buffer GA. Add 20 μl Proteinase K and 220 μl Buffer GB, mix, then incubate at 70℃ for 10 min until the solution clears. Add 220 μl ethanol, mix, and load the mixture onto the CB3 spin column. Centrifuge (12,000 rpm, 30 sec) and discard flow-through. Wash the column sequentially with 500 μl Buffer GD and twice with 600 μl Buffer PW (centrifuging 30 sec each time). Dry the column by centrifugation (2 min) and air-drying (2–5 min). Elute DNA with 50–200 μl Buffer TE (centrifuge 2 min). DNA concentration and quality were assessed using the NanoDrop (Thermo Fisher Scientific), and samples were stored at –20℃ until downstream analysis.” in Line 108-119
  • We have added the relevant sentence “PCR amplification of seven housekeeping genes was conducted in 50 μL reaction volumes containing: 25 μL of 2× Taq Master Mix (Vazyme, Nanjing), 2 μL of each primer (10 μM), 2 μL template DNA (~50 ng), and 19 μL nuclease-free water. Thermocycling conditions comprised initial denaturation at 95℃ (5 min); 30 cycles of 95℃ (30 s), 55℃ (30 s), 72℃ (1 min); and final extension at 72℃ (10 min). Amplicons were verified by 1.0% agarose gel electrophoresis, purified, and bidirectionally sequenced. Sequence types (STs) were assigned using the Salmonella MLST database (EnteroBase), and phylogenetic relationships were analyzed in GrapeTree v1.5.0 to generate a minimum spanning tree based on allelic profiles.” in Section “2.3 Multi-locus sequence typing of Salmonella isolates” in Line 123-130.
  • We have added the relevant sentence “For agar dilution, Mueller-Hinton agar (Oxoid) plates were prepared with serial two-fold dilutions of antimicrobials. Antimicrobial stocks were prepared following CLSI (2023) recommendations and stored at -80℃ prior to incorporation into agar. Bacterial suspensions adjusted to 0.5 McFarland standard (1–5 × 10⁸ CFU/mL) were diluted 1:10 in sterile saline to achieve 1–5 × 10⁴ CFU/spot. Suspensions were inoculated onto plates and incubated at 37℃ for 16–20 hr. Broth microdilution utilized cation-adjusted Mueller-Hinton broth (Oxoid) in commercially pre-coated 96-well plates (Thermo Fisher). Inocula were standardized to 0.5 McFarland (1–5 × 10⁸ CFU/mL) in saline, then diluted 1:100 in broth to achieve 5 × 10⁵ CFU/mL. A volume of 100 μL was dispensed per well. Plates were incubated at 37℃ for 16–20 hours.” in Section “2.4 Antimicrobial Susceptibility Testing (AST) of Salmonella isolates” in Line 138-160.
  • We have added the relevant sentence “Overnight cultures were washed in phosphate-buffered saline (PBS), adjusted to OD600= 0.5 (~1 × 108 CFU/mL), and mixed at donor:recipient ratios of 1:1 and 1:10. Cell mixtures (200 μL) were deposited onto 0.22-μm nitrocellulose filters placed on Mueller-Hinton agar and incubated at 37℃ for 18 hours. Following mating, the transconjugants ” in Section “2.7 Conjugation experiments of the mcr-1 gene” in Line 185-189.

Question 6: The Authors conclude that the salmonella levels in Chinese pork products were very high in 2023, nonethelss also more recent data would be needed to give a wider overall picture of this severe issue. Morevoer a comparison with similar situations, if observed in other Countries, would be useful to give a better assessment of the problem and to counteract it.

Response: We thank you very much for your valuable comments. We agree that ongoing monitoring for more recent data and international comparisons are crucial for a comprehensive understanding. Directly addressing the suggestion for comparison: Our study found a high Salmonella prevalence of 20.1% in Guangdong pig slaughterhouses. This rate is higher than domestic findings (e.g., 8.2% in Tibet) and substantially exceeds international rates (Portugal 2.0%, South Korea 9.1%, Brazil 2.9%, Sardinia 13.1%).

We have added the relevant sentence “This prevalence rate is higher than rates reported both domestically and internationally. Domestically, it exceeds the rate of 8.2% found in Tibet, China. Internationally, it is substantially higher than rates reported for Portugal (2.0%), South Korea (9.1%), Brazil (2.9%), and Sardinia (13.1%).” in Line 423-426.

Question 7: At the same time, in the Conclusion section, a perspective possible way to find a solution or possible alternatives should be added by the Authors after the data which they reported.

Response: This study reveals alarmingly high AMR levels in Salmonella from Chinese pork. To combat the threat, we recommend: (1) Strictly reducing and phasing out the use of colistin and other critically important antimicrobials in swine production to reduce selection pressure; (2) Accelerating development/deployment of alternatives like vaccines, probiotics, bacteriophages, and improved biosecurity; (3) Exploring novel interventions targeting epidemic plasmid spread; (4) Strengthening integrated "One Health" surveillance across human, veterinary, and environmental sectors along the pork production continuum.

We have added the relevant sentence “Building on these concerning findings, a multi-faceted mitigation strategy is imperative. This includes strictly reducing and phasing out the use of colistin and other critically important antimicrobials as growth promoters or prophylactics in swine production to alleviate selection pressure. Concurrently, accelerating the development and deployment of effective alternatives—such as vaccines targeting prevalent Salmonella serovars, probiotics, bacteriophages, enhanced biosecurity, and improved husbandry practices—is crucial to decrease reliance on antimicrobials. Furthermore, exploring novel interventions targeting epidemic plasmid dissemination, like conjugation inhibitors or CRISPR-Cas-based plasmid curing, could help limit the horizontal spread of mcr-1 and other resistance determinants. Strengthening integrated "One Health" surveillance systems across human, veterinary, and environmental sectors is essential for real-time detection of emerging resistance and enabling targeted interventions at critical points along the pork production continuum (farm, slaughter, processing, retail).” in Line 514-524.

Reviewer 3 Report

Comments and Suggestions for Authors

In this work the Authors investigated the prevalence and antimicrobial resistance of Salmonella isolates obtained from pork products in Guangdong province, China. Totally the Authors tested 457 pork samples collected in 2023 from two slaughterhouses. Salmonella isolates were found in 20.1% of samples, showing high resistance rates to tetracycline (90.2%) and multidrug resistance (58.7%). All Salmonella isolates were tested for mcr gene variants. The mcr-1 gene (encoding resistance to colistin), located on transferable Incl2 plasmids, was found in two S. Kentucky ST 198 isolates. Additionally in these isolates, using genomic analysis, the Authors identified a novel multidrug resistance region (MRR) and a variant Salmonella Genomic Island 1 (SGI1-KI) containing multiple antimicrobial resistance genes, which enhance Salmonella isolates survival under antibiotic selection pressure.

In my opinion the results showing the acquisition of complex resistance determinants (MRR and SGI1-KI) by S. Kentucky isolates are the most essential and original part of this work.

Generally this work is interesting and provides novel data to the existing knowledge, important for public health authorities and food producers. The manuscript is clearly presented and the interpretation is consistent with the obtained results. 

Minor concerns:

- chapter “Sample collection and Salmonella isolation” – add in this place (not only in Table 1) how many samples were tested from slaughterhouse 1 and 2

- give information how often the samples were taken (each week, month,...?) and how many during one sampling?

- what kind of pork products were tested and when were taken (before or after cooling of pig carcasses?)

Author Response

Reviewer 3

Comments to the Author

In this work the Authors investigated the prevalence and antimicrobial resistance of Salmonella isolates obtained from pork products in Guangdong province, China. Totally the Authors tested 457 pork samples collected in 2023 from two slaughterhouses. Salmonella isolates were found in 20.1% of samples, showing high resistance rates to tetracycline (90.2%) and multidrug resistance (58.7%). All Salmonella isolates were tested for mcr gene variants. The mcr-1 gene (encoding resistance to colistin), located on transferable Incl2 plasmids, was found in two S. Kentucky ST 198 isolates. Additionally in these isolates, using genomic analysis, the Authors identified a novel multidrug resistance region (MRR) and a variant Salmonella Genomic Island 1 (SGI1-KI) containing multiple antimicrobial resistance genes, which enhance Salmonella isolates survival under antibiotic selection pressure.

In my opinion the results showing the acquisition of complex resistance determinants (MRR and SGI1-KI) by S. Kentucky isolates are the most essential and original part of this work.

Generally, this work is interesting and provides novel data to the existing knowledge, important for public health authorities and food producers. The manuscript is clearly presented and the interpretation is consistent with the obtained results.

Response: Thank you very much for the positive comments, helpful suggestions and detailed corrections. We have reviewed the comments carefully and have revised the manuscript as your suggestions.

Minor concerns:

Question: (1) chapter “Sample collection and Salmonella isolation” – add in this place (not only in Table 1) how many samples were tested from slaughterhouse 1 and 2

(2) give information how often the samples were taken (each week, month,...?) and how many during one sampling?

(3) what kind of pork products were tested and when were taken (before or after cooling of pig carcasses?)

Response: We sincerely appreciate your question and valuable suggestions. We will address your questions point by point as follows:

(1) About “how many samples were tested from slaughterhouse 1 and 2”. The answer is “Between January 2023 and December 2023, a total of 457 pig carcass samples were collected from two slaughterhouses in Guangdong Province, China. 206 samples were collected from Slaughterhouse 1, and 251 samples were collected from Slaughterhouse 2.

(2) About “how often the samples were taken (each week, month,...?) and how many during one sampling?”

The answer is “Samples were collected monthly during eight specific months of 2023 (January, March, May, June, July, August, October, and December). During each monthly sampling event, a total of 57 carcass samples were typically collected from both slaughterhouses combined. The only exception was August, where 58 samples were collected. The samples per slaughterhouse varied slightly: Slaughterhouse 1 consistently provided 26 samples per month except in October and December (25 samples), while Slaughterhouse 2 typically provided 31 samples per month except in August, October, and December (32 samples). This monthly sampling design resulted in consistent totals across most months, with sampling occurring once per month over the study period”

(3) About “what kind of pork products were tested and when were taken (before or after cooling of pig carcasses?)

The answer is “The tested pork products were whole pig carcasses. Samples were collected immediately after slaughter but before the carcasses entered the cooling process. This pre-cooling timing ensures the samples reflect the initial microbial contamination level on the carcass surface prior to any temperature reduction or further handling. Each carcass sample was aseptically collected, sealed, and chilled for transport to the laboratory within 8 hours.”

We have added the relevant sentence “Between January 2023 and December 2023, a total of 457 pig carcass samples were collected from two slaughterhouses (Slaughterhouse 1 and Slaughterhouse 2) in Guangdong Province, China. This study was conducted with formal approval from the operational management departments of both participating slaughterhouses. Sampling was conducted across eight months (January, March, May, June, July, August, October, December). The monthly sample distribution per slaughterhouse was highly consistent: Slaughterhouse 1 contributed 206 samples (typically 26 samples/month, reducing to 25 in October and December), while Slaughterhouse 2 contributed 251 samples (typically 31 samples/month, increasing to 32 in August, October, and December). This resulted in a consistent total of 57 samples per month for most months, except August which had 58 samples (Table S1). Salmonella isolation was performed using a modified adaptation of the ISO 6579-1:2017 protocol. Pork products (carcass samples) were taken before cooling of the pig carcasses. Each sample was randomly and aseptically collected, sealed in sterile bags, and immediately stored in portable cooling boxes at 4–8℃. All samples were transported to the laboratory within 8 hours post-collection for processing. Upon arrival, approximately 25 g of each sample was rinsed with 225 mL of buffered peptone water (Huankai, China) as the initial processing step.” in Line87-101.
